# Study of Strain Capacity for High-Strain Marine Pipe

**DOI:** 10.3390/ma15165793

**Published:** 2022-08-22

**Authors:** Kun Yang, Ting Sha, Xiongxiong Gao, Hongyuan Chen, Qiang Chi, Lingkang Ji

**Affiliations:** 1State Key Laboratory for Performance and Structural Safety of Oil Industry Equipment Materials, Tubular Goods Research Institute of CNPC, Xi’an 710077, China; 2Xi’an Chang Feng Research Institute of Mechanism and Electricity, Xi’an 710065, China

**Keywords:** high-strain marine pipe, crack growth driving force, crack growth resistance, strain capacity, curved wide plate test, full-size bending test

## Abstract

In this paper, a strain capacity assessment on high-strain marine pipe was carried out by comparing the crack growth driving force and the crack growth resistance. The crack growth driving force was given by FEA, the stress-stain relationship was given by a DIC tensile test, and the crack growth resistance was given by a single-edge notched tensile (SENT) test using a single-specimen flexibility method. The proposed approach was compared with the failure assessment curve and validated against full-scale tests with a girth weld notch. The results of the full-scale tests showed that the assessment method using FEA was more accurate and the result of the failure assessment curve assessment was more conservative.

## 1. Introduction

With the discovery of abundant offshore oil and gas resources in deep-sea areas and the rapid growth in the marine pipeline industry, new developments that enable high operating pressure, long-distance traverses, and large diameter piping are essential [1]. Due to the high-pressure design of marine pipeline, new requirements for the grade, diameter and thickness of pipes should be explored. High-strain capacity pipe is needed for operation in the marine environment [2,3,4], both at the construction stage and in service.

In the marine pipeline construction stage, reel barges are used for pipe-laying [5]. The axial load of the pipeline is added during the reel-laying process, and a compressive or tensile strain is developed. Moreover, bending of the pipeline will occur as waves move, and the axial load of the pipeline is generated by bending. A compressive strain is formed in the side of the bending pipeline and a tensile strain is formed at the back. The strain capacity of a pipeline is required to meet the needs of the service environment [6], which is the principle on which the strain-based pipeline design method is based. 

Usually, the deformation capacity of a welding joint is less than that of the pipe metal, and girth welds are weak points in the pipeline. Thus, the strain capacity of the pipeline is dependent on the welding joint [7]. It is important to predict and evaluate the strain capacity of the welding joint. 

The types of failure in welding joints of pipelines can be divided into fracture and plastic collapses [8]. To ensure the strain capacity of the pipeline, low-stress brittle fractures should first be avoided. Overmatching the strength of a weld joint is necessary to ensure that the deformation area mainly appears in the pipe; the strain capacity of the pipeline is dependent on the properties of the pipeline steel. Ductile fracture is one of the most important types of pipeline failure. A ductile fracture can occur even though the weld joint is of overmatching strength. The strain capacity of the pipeline is dependent on the critical axial load during ductile fracture, which is much smaller than that for plastic collapse failure with an overmatching strength weld joint. 

It is hard to avoid defect formation during the welding process, and cracks may form and grow near defects when loads are exceeded. The process of fracture involves a competitive relationship between the crack growth resistance and the driving force. The former is related to the material properties, and the latter is affected by the external load [9,10]. Therefore, the risks of failure for the pipeline can be evaluated by comparison, and it is important for the assessment to use accurate data on the crack growth resistance and driving force.

Crack growth resistance is related to the crack tip constraint conditions. Analysis of the crack tip constraint effect has revealed that the geometry, crack depth, and loading conditions of the specimen affect the experimentally obtained fracture toughness [11]. By analyzing experimental data, researchers have found that the toughness and crack growth resistance curves of the material are related to the geometry of the specimen [12]. According to published reports, the specimen’s fracture toughness decreases as the level of constraint rises [13]. The crack tip constraint level for single-edge notched bending (SENB) and compact tensile (CT) specimens with deep cracks is the largest, while that of single-edge notched tensile (SENT) test specimens is closest to the crack tip constraint level of the axial cracks of the tube.

The crack growth driving force, in terms of the crack tip opening displacement (CTOD) or the J-integral, can be obtained by finite element analysis (FEA) [14,15,16]. Various structural geometries (e.g., crack depth, mesh size and node selection), material properties (e.g., stress-strain behavior) and load condition (e.g., the pressure, tensile or compressive stress) play a great role in the FEA result produced.

In engineering practice, crack growth, particularly unstable crack propagation (breakage), is regarded as unacceptable, the safety of running pipes must be under control, and the residual life should be precisely calculable [9,17,18]. The basic parameters (including yield stress, toughness, flaw size and so on) are usually applied according to relevant standards (e.g., BS7910, API579 or R6) to obtain a conservative safety assessment [19,20,21]. However, an approach that compares the driving force and crack growth resistance can provide a more accurate assessment result and is applicable in many fields.

## 2. Pipe Material and Welding Procedure

In this paper, a type of high strain marine pipe is used to investigate deformation capacity by considering the relationship between the crack growth driving force and the crack growth resistance (the R-curve). 

The crack growth driving force curves are given by the FEA, and the crack growth resistance curves of the specimens cut from the pipe are provided by single-edge notched tensile (SENT) tests. Curved wide plate (CWP) tests and full-size bending tests are also carried out to verify the results of simulations. The results of the safety assessment curve method are then compared to the simulation. 

Two pipes from the same factory (with a similar property) were used for the tests. The outside diameter (OD) of the pipe (Pipe A) used was 559 mm, the thickness was 31.8 mm, and the length of the pipe for the full-size bending test was 10,000 mm. The defect was produced as a linear notch, the length of the notch (2c) was 50 mm, the depth (a) was 4 mm, the crack tip was 0.15 mm. The defect was made on the inside wall of the pipe along the heat-affected zone (HAZ) of the girth welding (in the middle of the pipe) by electrical discharge machining (EDM), as shown in Figure 1a.

The specimens for the CWP, SENT and tensile tests were cut from another pipe (Pipe B) with the same welding technology (fully automatic welding). The specimens for the CWP test were machined from the weld joint along the longitudinal direction of Pipe B to obtain the critical tensile strain during failure; the size and the positions of the extensometers are shown in Figure 1b.

In Figure 1b, W = 275 mm, L = 1000 mm, LT = 1300 mm, and WT = 400 mm. The notch was made on the inside wall of the pipe along the heat-affected zone by EDM. Three size notches of CWP specimens were produced, which are shown in Table 1.

A full-size bending test was carried out using a larger four-points bending device. The length of the pipe was 10,000 mm. The pipe ends were held up by two pivots; the distance between the two pivots was 9000 mm. The notch was made on the inside wall of the pipe along the heat-affected zone by EDM; the size of the notch is shown in Table 1.

The specimens for the SENT test were machined from the weld joint along the longitudinal direction of the pipe to obtain the crack growth resistance. The stress-strain curve of the tube used for the FEA was given by a smooth bar tensile test with DIC (digital image correlation). The information on the SENT test (Figure 1c), tensile test (Figure 1d), curved wide plate and full-size bending test is shown in Table 1, and the chemical compositions of Pipes A and pipe B are listed in Table 2. Moreover, three specimens were used in the general mechanical tests (such as the tensile test and SENT) and the tests were reproducible.

## 3. Experimental Research

Mechanical performance tests were carried out to determine the material proprieties for use in the FEA process, including the tensile test by DIC (digital image correlation) and the SENT test.

### 3.1. Tensile Test

To obtain the true stress-strain curves of the pipe, tensile tests for the pipe metal with DIC and all-weld-metal using the traditional method (use extensometer) were carried out using an MTS universal testing machine, as shown in Figure 2. The MTS machine was powered by a hydraulic drive with a maximum load of 25,000 kg. Compared to the traditional method, the tensile test with DIC can obtain the stress-strain information during necking, and the material constitutive model derived from the stress-strain curve is better.

In the tensile test, the speckle was sprayed on the specimens, the coordinate information in the space of the speckle was recorded by two cameras when the load increased (Figure 2b), the strain cloud was calculated using the coordinate information, and the true stress-strain curves were finally obtained.

In the tensile test, the stress-strain curve (material properties) of the specimen was obtained by DIC observation (Figure 3), transformed into a true stress-strain curve and used in the FE models for different crack sizes.

Equations (1) and (2) provide the transformation of the true stress-strain curve.
(1)$$\sigma^{\prime }=\sigma \left(1+\varepsilon \right)\;\;$$
(2)$$\varepsilon^{\prime }=\ln\left(1+\varepsilon \right)$$
$\mathrm{where}\;\sigma^{\prime }$ is the true stress, $\varepsilon^{\prime }$ is the true strain, σ is the engineering stress, and ε is the engineering strain. The engineering stress and engineering strain can be obtained from Figure 3a.

In Figure 3a, the maximum load of the specimen appears at the end of stage 1. The maximum strain area occurs in the middle of the bar, which is shown in Figure 3b. However, necking did not happen at this stage. At the end of stage 2, the necking of the specimen occurred in the middle of the bar (see Figure 3c). It is difficult to record the whole strain condition between stage 1 and stage 2 by traditional measurement with an extensometer. Therefore, more stress-strain data is provided by the tensile test with DIC, and the true stress-strain curve used in the FEA is more accurate.

In Figure 4, the tensile stress and yield stress for the all-weld-metal are shown to be higher than for the pipe metal. However, the elongation for the pipe metal was higher. The tensile test by DIC gives more stress-strain data. Because of the welding process, the properties of the heat-affected zone (HAZ) are different from that of the pipe metal. 

Hardness data was obtained using a Vickers hardness tester(Kbprueftechnik, German). The indenter of the Vickers hardness tester was pressed vertically into the surface of the sample with a load of 0.5 kg. Figure 5a illustrates the 5400 hardness points (90 × 60, the space between two points is 0.5 mm) measured by the microhardness tester. The hardness cloud map (Figure 5b) was drafted using the hardness values. The hardness cloud map for the girth weld shows that the values of hardness in HAZ were softer than for the weld metal and pipe metal, especially in the area of root welding. Crack initiation usually occurs in the HAZ of the root welding. Safety evaluations often use a SENT test (the notch is made in the HAZ of root welding, Figure 1c) or CWP (the notch is made in the HAZ of the pipe inner surface, Figure 1b) to investigate the properties of pipes.

The properties of HAZ are similar to that of the pipe metal and difficult to obtain (HAZ is too small to test). The relationship between the tensile property and hardness is linear [22]. Thus, the stress-strain curves of HAZ can be obtained using the test results for the pipe metal according to the results of the hardness cloud map. 

The average value of hardness in the HAZ of root welding was 225 HV_0.5_, for the neighboring pipe metal was 237 HV_0.5_, and the soften ratio was 0.95. This ratio can be used to determine the true stress-strain curve for HAZ, which is shown in Figure 4. 

### 3.2. SENT Test

The crack growth resistance curve was derived from SENT test results. A single-specimen flexibility method with two identical MTS extensometers was used in the tests. The gauge length of each extensometer was 5 mm, and the maximum measurement range was 10 mm. The extensometers were installed on a tooling setup (Figure 6a,b). The crack open displacement results were recorded for the lower extensometer (V1) and the other extensometer (V2) with increasing load. The CTOD was then calculated using the geometrical relationship between V1 and V2, and the curve showing the relationship between CTOD and the crack extension (crack growth resistance curve) was obtained (see Figure 7).

The fitting function in Figure 7 is
(3)$$y=ax^{b}\;\;\;$$
where *a* = 1.52 and *b* = 0.42.

## 4. FEA Process

The FE models of the whole pipe with four crack sizes were constructed using Abaqus 6.11 (Dassault Systèmes, France). The four notch sizes (depth × length) were 4 mm × 50 mm, 7 mm × 87.5 mm, 10 mm × 125 mm, 13 mm × 162.5 mm and 16 mm × 200 mm, respectively. The length of the pipe was 12,000 mm, the thickness was 31.8 mm, the original CTOD was 0.2 mm and the OD was 559 mm, which was the same size as the pipe in the full-size bending test.

The welding joint was divided into pipe metal, HAZ and weld metal. The color of the meshes were white, green, blue and red, as shown in Figure 8b. The material properties (true stress-strain curves) of the pipe metal, HAZ and the weld metal were assessed. The crack was set up between the HAZ and weld metal (along the fusion line), as shown in Figure 8c.

Fine mesh was employed in the crack tip regions, as large plastic deformation occurs when the load is applied. In order to preferably simulate the passivation process of the crack, mesh refinement was used in this area and circular arc transition was adopted near the end of the notch, see Figure 8c,d. This crack simulation method has been shown to reduce mesh distortion and stress concentration effectively, which can simulate the crack tip passivation behavior during the crack opening process, so the corresponding fracture parameters (CTOD) are obtained accurately. Other areas of the model are divided by relatively coarse mesh. C3D8R elements are used to improve the efficiency of computation. 

The whole process was divided into two parts. Firstly, external pressure was added to the pipe to simulate the effect of sea pressure. When the pressure value rose to 15 MPa (1500 m depth of the sea), tensile stress was applied along the axial direction of the pipe.

In the second step, the tensile stress increased with time, and the CTOD value for the tube became larger at a higher strain, see Figure 9. 

The CTOD values were obtained by measuring the distance between the two nodes (CTOD1, CTOD2 and CTOD3) near the crack tip. CTOD1 is the distance between the two nodes on the edge of the circular arc transition mesh, CTOD3 is the value a little further than CTOD1, and CTOD2 is the middle value, as shown in Figure 9a. 

The three values of CTOD (CTOD1, CTOD2 and CTOD3) are equal when the tensile strain is 0. However, the three values were different and CTOD3 was the largest, with a tensile strain of 0.019. In the position of CTOD1, the mesh distortion is sensitive, and is difficult to relate to the CTOD at the crack tip. Moreover, the position of CTOD3 is quite far from the crack tip, and the distance does not represent the CTOD. Thus, CTOD2 is preferred and can be used to correspond to the CTOD at the crack tip.

The relationship between CTOD and strain was recorded and the data was fitted. The curves for the crack driving force with different notch sizes are plotted in Figure 10.

As shown in Figure 10, the CTOD values were 0.15 mm (notch depth was 4 mm), 0.3 mm (notch depth was 7 mm) and 0.95 mm (notch depth was 10 mm), and 2.0 mm (notch depth was 13 mm) for the pipe when the tensile strain was 0.006. Based on the shallowest crack depth tube (crack depth was 4 mm), the crack extension of the pipe with 7 mm crack depth was 3 mm, the crack extension of the pipe with 10 mm crack depth was 6 mm, the crack extension of the tube with 13 mm crack depth was 9 mm, and, of course, the crack extension for the tube with 4 mm crack depth was 0 mm. Using the CTOD values and the corresponding crack extension values, the crack growth driving force curves with different tensile strain values were obtained..

In Figure 11, the crack growth driving force curves with different tensile strain values are shown. When the crack depth of the pipe is less than 9 mm, the focus between the crack growth resistance curve and the crack driving force curves will not appear when the tensile strain is less than 0.01. This means that the crack growth resistance is higher than the driving force, and the crack will not propagate until the crack depth becomes larger. Moreover, there is a curve focus when the strain is 0.015. The crack depth is 6.9 mm and the CTOD value is 3.4 mm (pointed out with a red arrow in Figure 11) when the strain is 0.015, and the crack growth driving force is equal to the resistance. When the tensile strain is higher than 0.015, or the crack depth is larger than 6.9 mm, the crack growth driving force is always higher than the resistance, and the crack will run and not stop until it penetrates the whole wall. Then breakage of the weld of the pipe by crack propagation may finally occur. 

Using a non-destructive testing (NDT) method, it is easy to detect risk based on when the crack size of the pipe has a larger value. In the project, the height of a weld bead (4 mm) was considered the minimum size of NDT. In Figure 11, there is a curve focus when the strain is 0.04 and the crack depth is 4 mm (pointed out with a black arrow in Figure 11). The maximum tensile strain of the pipe is 0.04.

For comparison with the results of the FEA method, the failure assessments of the pipe with a crack were carried out using BS7910 and R6, as shown in Figure 12. The two failure assessment curves were close. The pipe was safe as the tensile strain observed was 0.01 and 0.025 according to BS7910 and R6, respectively. Moreover, the point is at the right side of the failure assessment curve as the tensile strain is 0.04; this implies that pipe breakage (weld failure) will occur, creating a significant risk.

The results of the failure assessment curve are more conservative than the FEA method. As the crack depth is 4 mm and the tensile is less than or equal to 0.04, the assessment results of the FEA method are always safe; however, the assessment result of BS7910 is at risk when the strain is 0.04.

## 5. Curved Wide Plate and Full-Size Bending Test

To verify the assessment results, curved wide plate and full-size bending tests were carried out. In Figure 1b and Figure 13, LVDT1 and LVDT2 give the global longitudinal strain of the specimen. Moreover, the remote strain is given by LVDT3 and LVDT4. The pre-existing crack in the CWP specimen does not run through the weld joint during the tensile load, and the failure model is the necking of the pipe metal, which is far from the weld joint. This indicates that weld failure will not occur during the tensile load when the initial crack depth is 4 mm; the tensile strain of CWP is dependent on the properties of the pipe metal (plastic collapse), and the predicted result of the FEA agrees well with the test.

In Table 3, the results for the CWP specimens are given. The failure model is pipe metal necking, as the initial crack depth is less than 4 mm, and the remote tensile strain is due to the properties of the pipe metal. The critical remote tensile strains of NO. 1 and NO. 2 are 5.00% and 4.98%, respectively. The crack runs through the whole wall of the CWP specimen by the tensile load when the initial crack depth is 8 mm (NO. 3) and the critical remote tensile strain is 2.14. In Figure 11, the tensile strain is less than 2.0% and greater than 1.5% when the initial crack depth is 8 mm. Compared to the CWP test, the assessment by FEA represents a conservative result.

Compared to the CWP test, the service condition of the full-size bending test is much closer to the running pipeline. In Figure 14, a piece of equipment designed to finish the full-size bending test with the oil press machine is shown. This equipment is a larger four-points bending device. The maximum load of the oil press machine is 6,000,000 kg, and the movement of the machine indenter is directed by a displacement control device. The test will stop when the load drops.

The pipe ends are held up by two pivots, and the press load is applied in the middle part of the pipe in a 0° direction by two indenters, as shown in Figure 14a,b. The distance between the two pivots is 9000 mm, and the distance between the two indenters is 1118 mm. The 180° direction of the pipe is loaded by tensile stress and the 0° direction of the pipe is loaded by compressive stress as the pipe bends. Thus, the critical tensile strain is finally determined. 

If a 5% drop in load is observed, a crack runs through the pipe wall or a pipe rupture occurs, then the test is stopped. The remote tensile strain (an outside diameter length away from the girth weld) is measured by strain gauges at that time.

The test stops when the load drops by 5%, and the results of strain gauges are shown in Table 4. The tensile strain is distributed symmetrically from the girth weld to the end of the pipe. When the strain value of No. 1, No. 2, No. 7 and No. 8 (near the indenters) is higher, the strain value of No. 13 and No. 14 is lower, indicating that the deformation is mainly caused by tensile stress. The strain values are higher on the pipe surface around the indenters; the plastic deformation of the pipe metal occurs mainly in these areas.

The crack does not propagate during the loading, and the failure model is the pipe rupture. The assessment result of the FEA agrees well with the full-size bending test.

Figure 15 shows the crack growth driving force curves for different pressures with a crack depth of 13 mm. The crack driving force increases with increasing external pressure of the pipe. The depth of the sea helps to reduce the crack driving force and improve the safety of pipes, and the FEA assessment can produce reliable results for marine pipelines.

## 6. Conclusions

A pragmatic approach to the assessment of strain capacity for high-strain marine pipe, considering the crack growth driving force and the crack growth resistance, is proposed in this paper. An accurate assessment of crack propagation and critical tensile strain was carried using an FEA method, compared to the failure assessment curve, and verified by CWP and full-size bending tests. The assessment results showed that a pipe with a maximum allowable weld defect can cope with considerable remote strain. The following conclusions are drawn:-The FEA method results showed that the crack growth resistance curve intersected a driving force curve of 0.04 tensile strain with a crack depth of 4 mm, and that unstable crack propagation will occur when the crack depth is larger than 4 mm and the tensile strain is higher than 0.04. The marine depth helps to reduce the crack driving force and improve the safety of the pipeline, and the FEA assessment can produce reliable results.-The results of CWP tests indicated that crack running does not occur when the initial crack depth is less than 4 mm, the failure model is the rupture of pipe metal, and the critical strain capacity is 0.05. Fracture occurs when the initial crack depth is 8 mm and the critical strain capacity is 0.02. The crack does not run in the full-size bending test, and the failure model is the pipe rupture.-The assessment comparison results show that the result of the failure assessment curve is more conservative; the FEA method is preferred and agrees well with the full-scale test.

## Figures and Tables

**Figure 1 materials-15-05793-f001:**
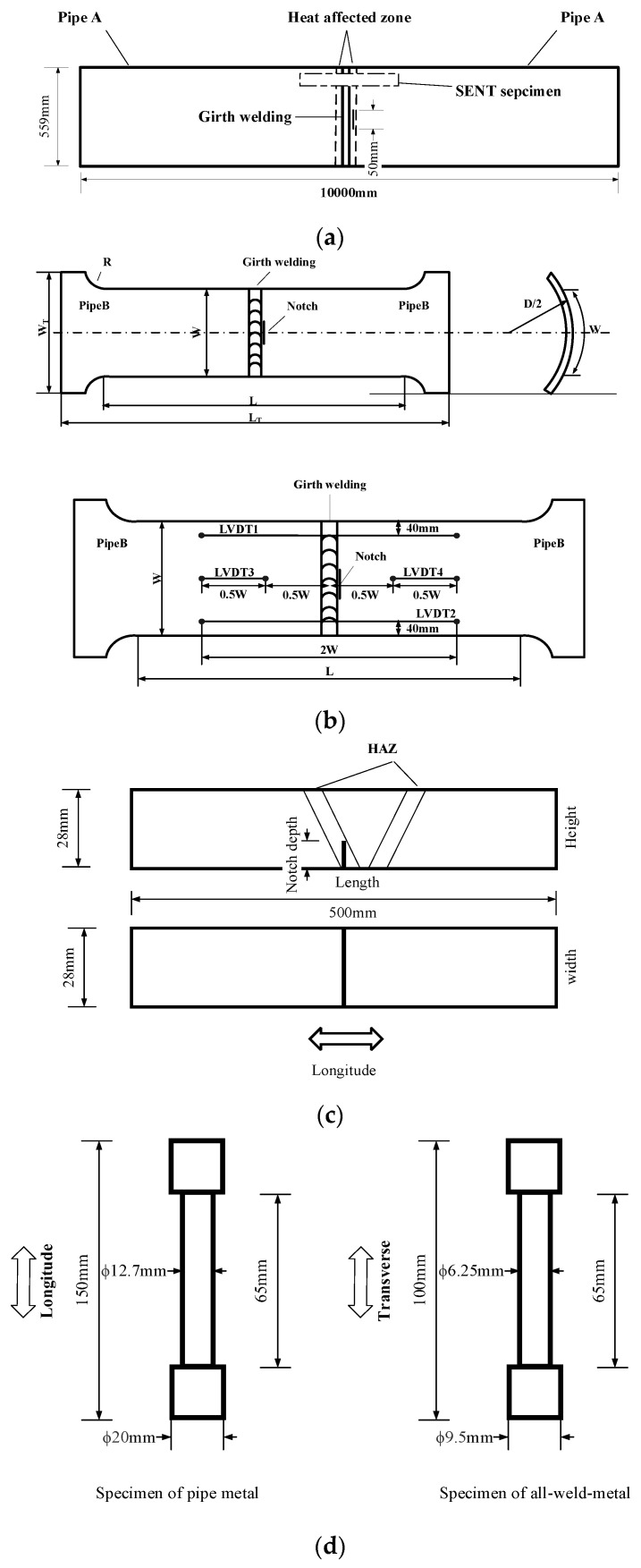
The size of SENT, tensile test and full-size hydrotest cuts from the high strain marine pipe. (**a**) The size of pipe used in the full-size bending test; (**b**) Specimen of CWP and the positions of extensometers; (**c**) Specimen of SENT test; (**d**) Tensile test specimens.

**Figure 2 materials-15-05793-f002:**
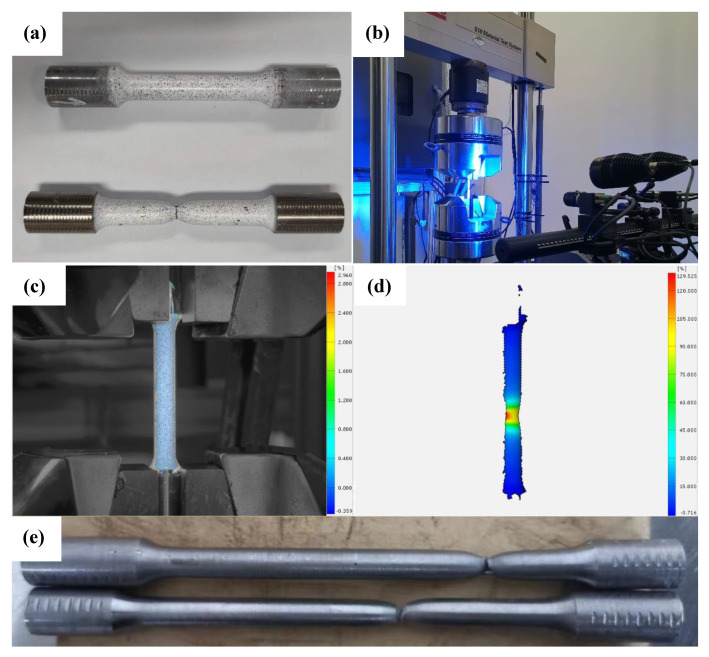
Tensile tests of pipe metal and all-weld-metal for the high-strain marine pipe. (**a**) The specimens of tensile test by DIC; (**b**) Tensile test of pipe metal by DIC; (**c**) Speckle area on the specimen of pipe metal; (**d**) Speckle analysis; (**e**) Tensile specimens of all-weld-metal.

**Figure 3 materials-15-05793-f003:**
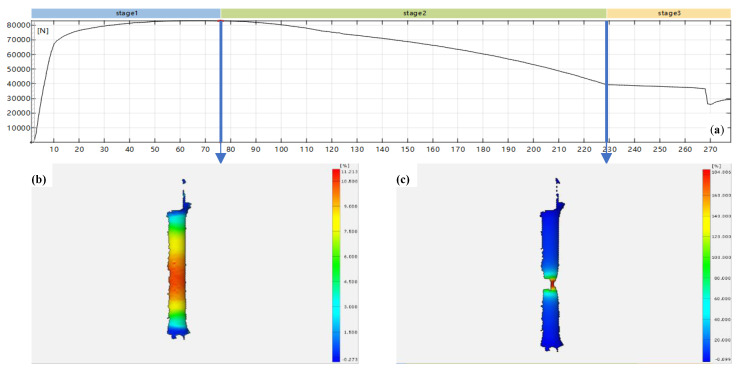
Tensile test result of pipe metal by DIC. (**a**) Load-number of image frames curves; (**b**) The cloud picture of tensile strain at the end of stage 1; (**c**) The cloud picture of tensile strain at the end of stage 2.

**Figure 4 materials-15-05793-f004:**
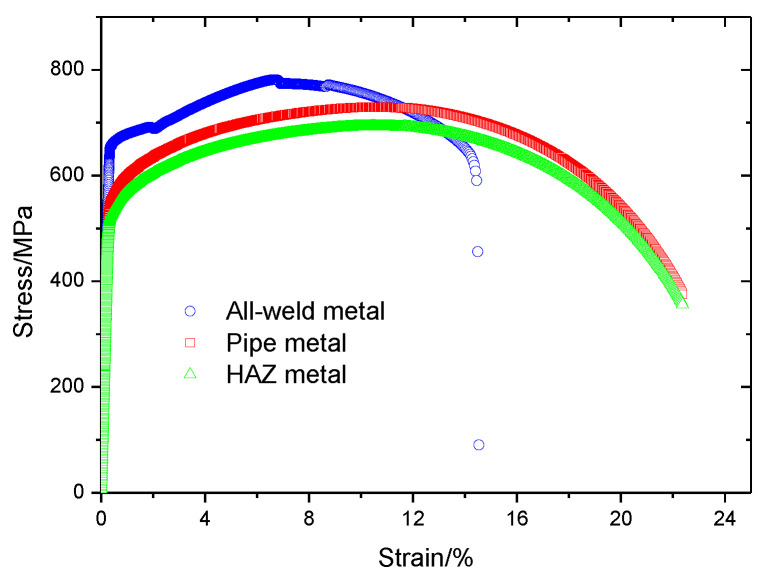
True stress-strain curve for pipe metal by DIC, HAZ metal and all-weld-metal.

**Figure 5 materials-15-05793-f005:**
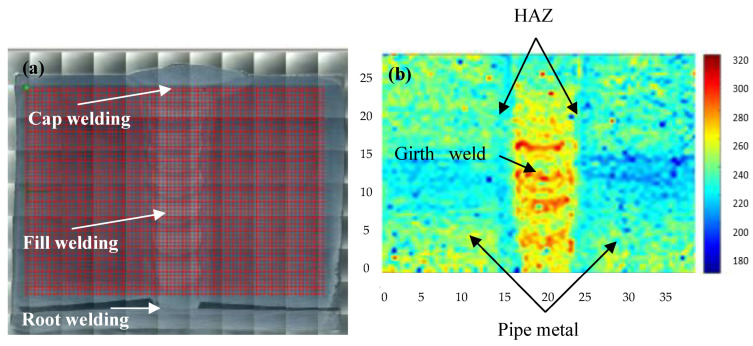
The hardness cloud map of girth welding joint for the high-strain marine pipe. (**a**) The zone of hardness cloud map for the girth welding joint; (**b**) The cloud map for the specimen.

**Figure 6 materials-15-05793-f006:**
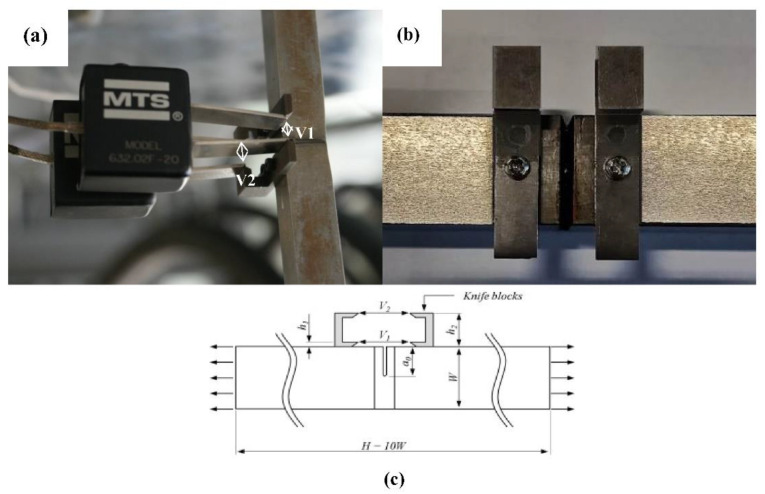
Single-edge notch tensile test. (**a**) Test with two MTS extensometers; (**b**) The tooling used to install the extensometers; (**c**) Schematic diagram of SENT.

**Figure 7 materials-15-05793-f007:**
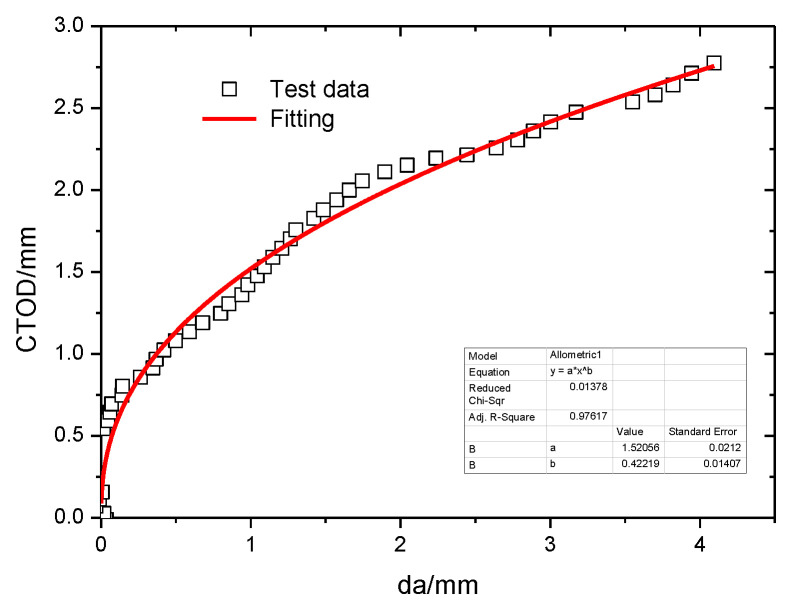
Crack growth resistance of the specimen (notch depth is 8.4 mm) by SENT.

**Figure 8 materials-15-05793-f008:**
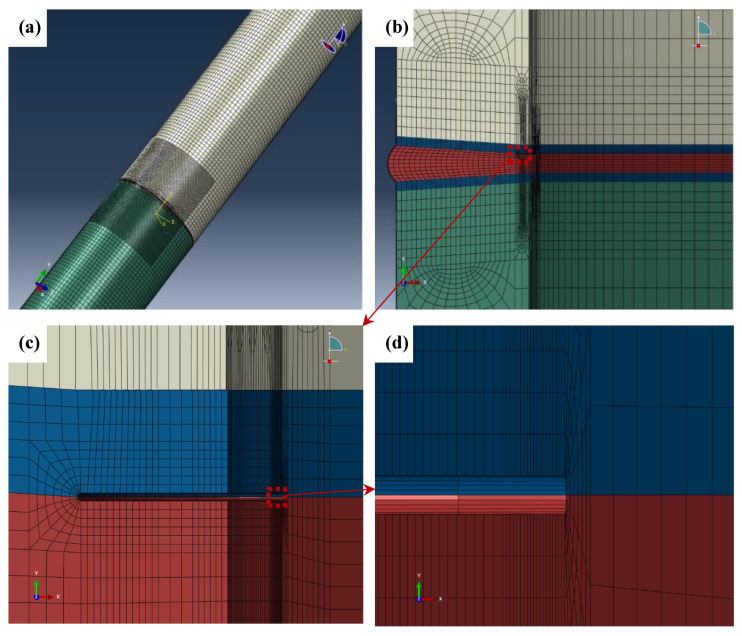
Schematic diagram of FE model for the tube (crack depth is 4 mm). (**a**) The pipe model; (**b**) Mesh of the welding joint; (**c**) Mesh of the area near the crack tip; (**d**) Mesh of the area near the end of the notch.

**Figure 9 materials-15-05793-f009:**
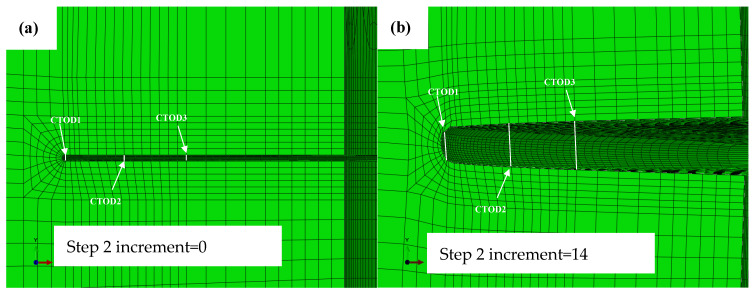
Crack tip mouth changes with load for the FE model (the notch depth is 7 mm). (**a**) The crack tip with tensile strain 0; (**b**) The crack tip with tensile strain 0.019.

**Figure 10 materials-15-05793-f010:**
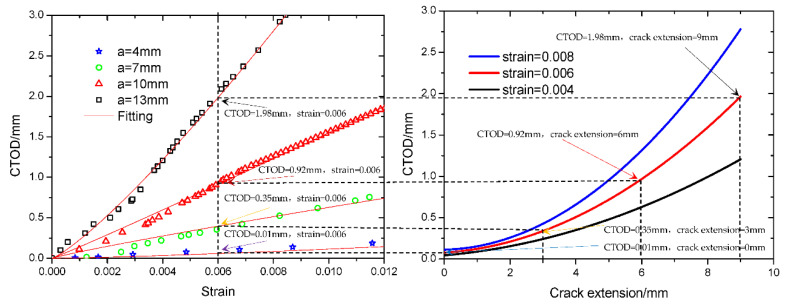
Crack growth driving force curves calculated using FEA results.

**Figure 11 materials-15-05793-f011:**
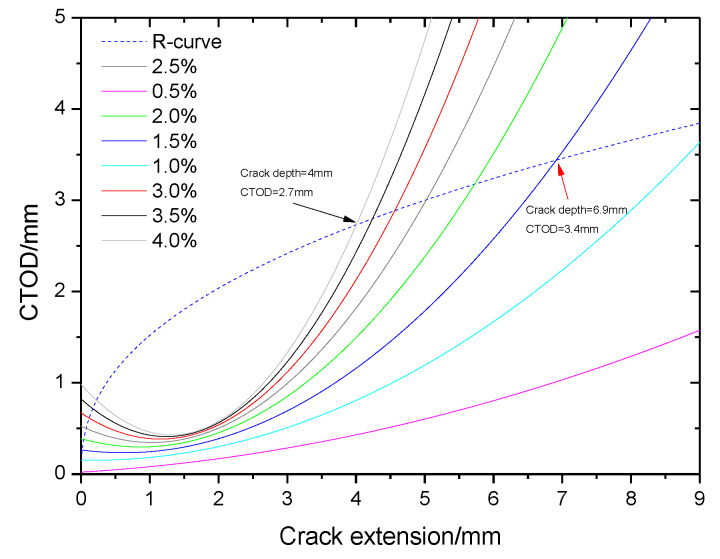
Comparison between crack growth resistance curve and crack growth driving force curves.

**Figure 12 materials-15-05793-f012:**
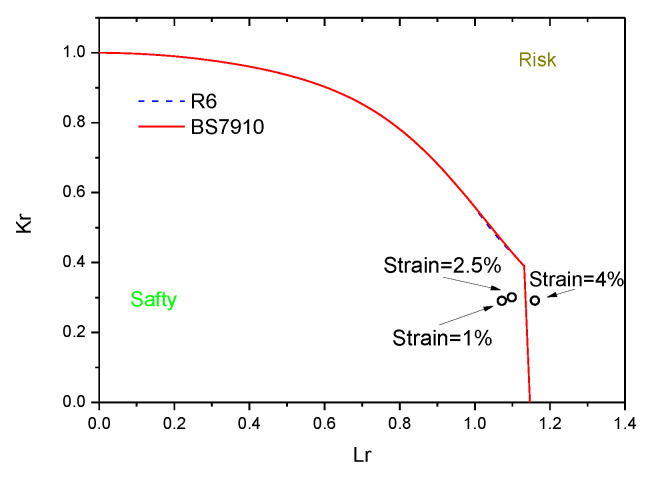
Failure assessment curve of the pipe by BS7910 and R6 with initial crack depth of 4 mm.

**Figure 13 materials-15-05793-f013:**
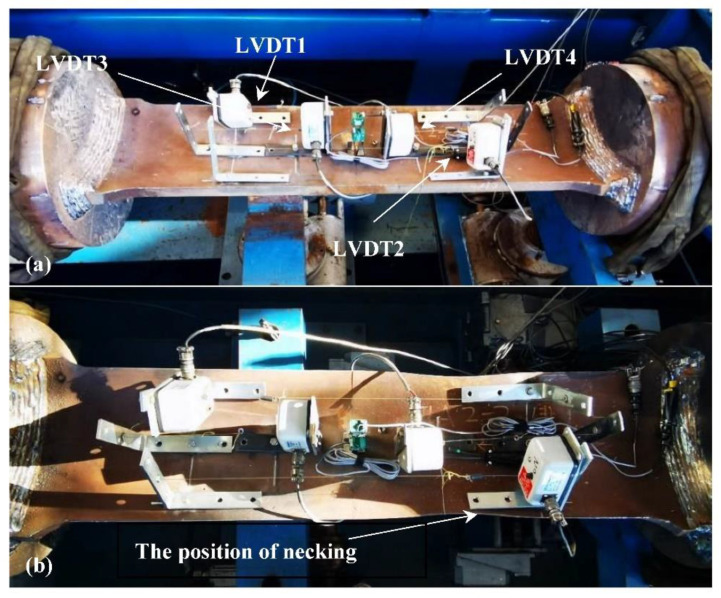
The test of curved wide plate (CWP) when the initial crack depth is 4 mm. (**a**) Extensometers installation on the specimen; (**b**) The position of necking after test.

**Figure 14 materials-15-05793-f014:**
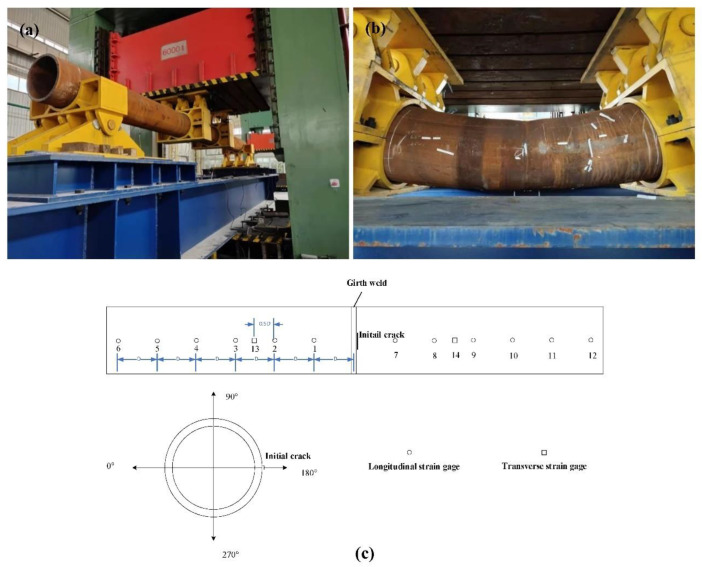
Full-size bending test for the high-strain marine pipe. (**a**) Test setup; (**b**) Bending area of the pipe; (**c**) Diagram of strain gauge positions on the pipe.

**Figure 15 materials-15-05793-f015:**
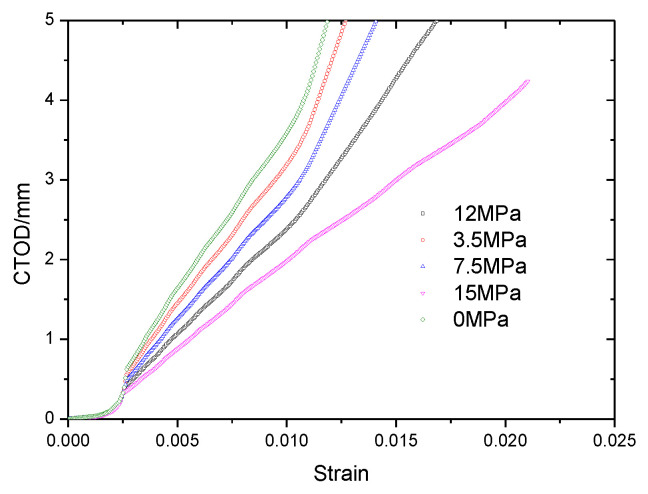
Crack growth driving force curves with different external pressure with a crack depth of 13 mm based on FEA results.

**Table 1 materials-15-05793-t001:** The specimen size of SENT, tensile test, CWP and full-size bending.

SENT	Length × Width × Height	Notch depth	Instrument and Standard
	500 mm × 28 mm × 28 mm	8.4 mm	MTS810; SY/T 7318.2-2016
Tensile test	Length × Diameter	/	SHT4106-SUNS; API 5L-2018
	150 mm × ϕ12.7 mm 100 mm × ϕ6.25 mm	/	
CWP	Length × wide × Thickness	Notch size (2c × a)	Combined Load Applying System; SY/T 7318.1-2016
	1000 mm × 275 mm × 31.8 mm	25 mm × 2 mm 50 mm × 4 mm 100 mm × 8 mm	
Full-size bending test	Length × Outside Diameter × Thickness	Notch size (2c × a)	SY∕T 7318.3-2017
	10,000 mm × 559 mm × 31.8 mm	50 mm × 4 mm	

**Table 2 materials-15-05793-t002:** Chemical compositions of Pipe A and Pipe B (Wt × 10^−2^).

	C	Si	Mn	P	S	Mo	Ti	N
Pipe A	0.049	0.16	1.74	0.0082	0.0024	0.096	0.011	0.0033
Pipe B	0.048	0.16	1.74	0.0087	0.0029	0.097	0.011	0.0034
	Cr	Nb	V	Ni	Cu	B	Al	Pcm
Pipe A	0.29	0.053	0.0051	0.10	0.033	0.0003	0.031	0.17
Pipe B	0.29	0.053	0.0055	0.10	0.034	0.0003	0.030	0.17

**Table 3 materials-15-05793-t003:** The results of the CWP test.

NO.	Crack Size (mm)		Remote Strain (%)			Failure Model
	Depth	Length	LVDT3	LVDT4	Average	
1	2	25	4.02	5.98	5.00	Necking
2	4	50	3.69	6.26	4.98	Necking
3	8	100	1.52	2.76	2.14	Fracture

**Table 4 materials-15-05793-t004:** The tensile strain result of strain gauges in Figure 14c during the maximum load.

Position	1	2	3	4	5	6	7	Failure model
Value	4.0%	4.5%	3.0%	2.1%	1.8%	0.8%	4.1%	pipe rupture
Position	8	9	10	11	12	13	14	
Value	4.5%	3.2%	2.0%	1.9%	0.9%	0.07%	0.08%	

## Data Availability

Some or all data, models, or code generated or used during the study are available from the corresponding author by request.

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
