# Peer review of "Study of Strain Capacity for High-Strain Marine Pipe"

_materials, 2022, doi:10.3390/ma15165793_

Round 1

Reviewer 1 Report

Title: Study on strain capacity for high strain marine pipe

Paper Id: materials – 1831298

Specific comments

1.      In line 30 – it is stated that axial load of the pipe line is produced by bending. How bending causes axial load?

2.      Why high strain capacity is needed only for marine environment? Is it not same issue with deep underground pipes in urban cities, metro pipelines, etc.?

3.      Line 45 - What is meant by plastic collapse failure?

4.      In line 44 to 46, it is stated that the strain capacity during critical axial loading is less than the failure condition during plastic collapse. Why it is so?

5.      Line 74 to 75 – it is said that comparison of crack growth resistance and driving force can give a more accurate assessment. Not explained. Also when you do this comparison at lab conditions, whether same situation can be extrapolated to marine condition, where environment, pressure flow, etc are different?

6.      Line 79 – be specific as resistance to what?

7.      L 100 – some explanation is required on hydrotest

8.      L 132 – what is materials constitutive and why the one obtained from stress strain curve with DIC is better? Afterall, it is obtained from the stress – strain data, and how image correction helps in it?

9.      L 169, Fig 4: Identification of 3 plots in the figure is missing.

10.  No explanation on welding procedure is given. Is it an optimised welding procedure for stated applications? Also, any NDT qualifications are done to ensure that the flaws are not dangerous.

11.  L 250 – From fig 11, crack growth curves, how the conclusion of 6.9 mm crack growth was observed when the strain is 0.015? Similarly conclusion of such numbers from the plots is not clear. Similarly fixation of strain as 0.006 in Fig 10. Similarly, how the conclusion was drawn as stated in line 247-248; 252-253?

12.  L 256- Crack is always an unwanted flaw. What do you understand by stating detection of risk?

13.  L 263 -  Fig. 12 – Two failure assessment curves are exactly overlapping (they are not only close!). Needs more explanation in the text.  

14.  L 266 – it is said that the point is at the right side of the failure assessment curve as the tensile strain is 0.04. Such critical points should be marked clearly, both in Fig 11 and Fig 12.

15.  Figure caption (14c ) and Table captions (table 3) are not correctly presented. Table 3: four rows are not identified.

16.  Figure 15 presents effect of external pressure affecting crack growth force. Still you claim -marine depth will reduce the crack driving force.

General comments

English grammar is poor. Many a time, difficult to understand. A few of them are listed.

L 30 – compress strain is not correct. It should be compressive strain.

Compress strain is formed -is not correct

L36 – dependent instead of depended. Same in L42

L40 – needed instead of need

L48 – load exceeded not load existed. There are many.

Reviewer 2 Report

The paper submitted for review is presented for investigation by depth FEA approach and experimental one of the strain capacity of the marine pipe.

The authors have processed a considerable amount of data, but the following remarks can be made about their work:

-        -The template of the journal must be downloaded and use to submit the article.

-       - Line 77 has a typo it is not “pf high…” it is “of high…”.

-       - It would be good to present the material from which the pipes used in the study are made - chemical composition, mechanical characteristics. Perhaps the authors had this in mind when they indicated table 2, but table 2 shows completely different results.

-       - It is necessary to describe the equipment used throughout the article – hardness tester, tensile testing machine, extensometer, 4-point bending machine.

-       - Line 132 – missing the word "model" after "material constitutive..."

-       - Fig. 3 represents the "Engineering Stress strain curve", in order to use it in FEA, it is necessary to present the way in which the transformation in the "true stress strain curve" was performed. In this sense, the sentence on line 142 is not formulated correctly.

-      -  To cite a suitable reference for the statement: "The relationship between tensile properties and hardness is linear."

-      -  The curves in Fig. 4 should be recalculated using the chosen methodology in "true stress strain"

-     -   It is necessary to present in the main text , the power function obtained after approximating the experimental points shown in Fig. 7.

-       - Line 198 – If the Constitutive model of material shown in Fig. 4 is used in FEA, then the results obtained after FE analysis must be recalculated, due to my previous remarks.

Round 2

Reviewer 1 Report

Title: Study on strain capacity for high strain marine pipe

No: materials-1831298-peer-review-v2

One of the comments during previous review is extensive correction of English language. This aspect is not done. I have listed few corrections as examples. Only listed errors are corrected. The paper has vast number of language corrections. I have listed them, pertaining to first half page. Because, the reviewer is not for correcting the grammers. of course he can highlight, if it is a few.

Technically there is substantial improvement.

Following are the corrections in the first half. 

Page 1, line 11: correct as resistance for crack growth

Line 14:  correct as compared with failure assessment curve

Line 15: correct as the assessment method using FEA is

Line 21: correct as With the discovery of the abundant

Line 23 Correct as large diameter piping are essential

Line 25: Correct as for pipes should be explored.

Line 28: correct as are used for pipe laying

Line 30: Correct as tensile strain is developed

Line 31: correct as waves move, P10/17

Introduce a space between the value and unit   

Reviewer 2 Report

For the most part of the paper, the remarks made have been corrected. It remains:

On line 177 - When measuring hardness, it is unnecessary to specify that the applied load is compressive.
